# Evaluation of Happy Sport, an Emotional Education Program for Assertive Conflict Resolution in Sports

**DOI:** 10.3390/ijerph19052596

**Published:** 2022-02-23

**Authors:** Agnès Ros-Morente, Miriam Farré, Carla Quesada-Pallarès, Gemma Filella

**Affiliations:** 1Department of Pedagogy, Universitat de Lleida, 25001 Lleida, Spain; mfarresopena@gmail.com (M.F.); gemma.filella@udl.cat (G.F.); 2Serra Hunter Fellow, Department of Applied Pedagogy, Autonomous University of Barcelona, Campus Bellaterra, 08193 Cerdanyola del Vallès, Spain; carlaquesada@gmail.com

**Keywords:** gamified emotional education program for sports, coexistence, video games, conflicts

## Abstract

Background: Interpersonal conflicts occur in any kind of social relation, including the field of sports. Proper emotional management can improve athletes’ well-being, coexistence, and performance. This study presents the initial results of the gamified emotional education program Happy Sport in a sample of athletes in the field of non-formal education. Methods: The study sample consists of 194 athletes from the *benjamín* and *alevín* categories (3rd- to 6th-grade primary school children). A quasi-experimental pre-intervention and post-intervention design with a control group is followed using the Games and Emotions Scale (GES), Social Support Scale, Emotion Regulation Questionnaire for Children and Adolescents (ERQ-CA), and Bullying in Sports Questionnaire. Results: Statistically significant differences were found across participants in the experimental group between the pre- and post-intervention evaluations for the variables satisfaction and bullying. An analysis of the competencies related to emotion regulation revealed significant results for the experimental group for both scales (cognitive reappraisal and expressive suppression). Conclusions: The results show that after a training session with the gamified software Happy Sport, children’s satisfaction increased and bullying levels decreased. Changes in cognitive reappraisal and expressive suppression may also be explained by the training received.

## 1. Introduction

Conflicts can occur in any type of human interaction and sports are no exception [1]. Sports contexts facilitate coexistence between teams, training groups, etc., which can sometimes involve a certain amount of conflict between athletes [2].

Contrary to popular belief, studies on the specific characteristics of peer conflict have established that, as noted by [3], many youth aggressions toward peers are not due to an excess of hostility but rather a lack of skills and strategies for solving social problems effectively. In fact, many studies have linked conflict-prone behaviors to a lack of emotional competencies [4,5,6,7,8,9,10,11]. A low perception of emotion regulation has also been found to be associated with a high percentage of negative behaviors, a low frequency of positive behaviors, and a high rate of bullying [12,13,14,15].

In this regard, numerous studies have shown that children’s and adolescents’ ability to manage their emotions appropriately is a condition that guarantees success in fostering interpersonal relationships, coping with adverse situations, goal achievement, and, in general, individual psychological adjustment [16,17,18,19].

Therefore, it seems only logical that, as some studies point out, cultivating and improving emotional competencies helps resolve and prevent the emergence of conflict in various settings, including sports [20]. Additionally, a growing body of research suggests that proper conflict management fosters well-being, strengthens relationships, and enhances performance [21,22]. Consequently, studies on emotion regulation and emotion management have become increasingly important in various fields of research, with sports being among them [23].

In the area of sports, authors such as [24] have also found that athletes improve their sports performance when there is more cohesion and emotional wellbeing on the team, a fact that enhances the enjoyment and commitment of each individual participating in the sport [25]. Similarly, other authors, such as [25], demonstrate the importance of inducing positive emotions in athletes in order to improve and facilitate adaptation and positive outcomes and, at the same time that, this triggers positive emotions that are essential for successfully coping with the systematic processes of training and/or competing. Contrary to that, when there is an increase of negative emotions, such as anxiety, there is a negative association with athletes’ sports performance, well-being, and health [26].

Therefore, one can conclude that emotions play a very important role in sports. This gives rise to the need for proper training in emotion management. This will provide the possibility for exhibiting better performance and better cohesion within teams [27].

In sports schools’ context so far, there is not a tradition of promoting emotional intelligence or training in emotional competencies. This can often lead to dissatisfaction among students, in addition to insecurity and a lack of clear values [28].

One reason why physical education largely neglects emotions today is probably because it is perceived as a discipline that is focused on the concept of movement, which is clearly defined and devised in terms of mechanical physics [29]. More recent studies, such as that by [30], propose focusing physical education on the study and development of the body in motion and understanding the body holistically, not only as a set of muscles, bones, joints, and their biomechanical connections, but as a body that thinks, feels, listens, communicates, expresses itself, etc., and that can only be conceived of based on the combination of these three dimensions: the mental (through thoughts), the emotional (through feelings and emotions), and the physical (through motor activity).

However, the development of strategies to regulate these and other emotions is considered essential, as it promotes concentration on the task at hand and reduces the anxiety felt toward the sport [31]. Such adaptive emotion regulation strategies should be implemented in a dispositional way during training and prior to competition, especially in youth sports [32]. Hence, the importance for athletes to develop psychological skills to positively channel competitive coping responses and make them a source of enjoyment [33].

For this reason, and based on all these suppositions and research, Happy Sport was thought of and created as an educational video game designed to cultivate emotional competencies for sports students. Players are presented with 25 conflicts that they must learn to resolve assertively with the help of training for emotional competencies. Thus, the overall objective of the video game is to improve the management of emotional competencies, reduce the number of conflicts, improve the game climate, and reduce anxiety, thereby resulting in enhanced well-being and performance for the team. Happy Sport is intended to be used with complementary material and (previously trained) trainers to guide the program and optimize its implementation.

When using the Happy Sport program, participants train their emotional competencies by solving and managing virtual conflicts that appear during the game.

After choosing their avatar, the participant finds a sport context in which he or she encounters different conflicts, which are represented by a symbol on the minimap. The presentation of the conflict appears and then there is a dialogue among different characters. The participant plays different roles: in some conflicts the participant will be a person who can be abusive or aggressive, in others the participant will be a person that is bullying, and on other occasions the participant can just act as an observer. After the presentation of the conflict, the following message appears in the game: “What emotions do you feel? You can select seven possible emotions”. Afterwards, there is the possibility of practicing emotional understanding and legitimizing the emotions. Finally, participants can choose from different regulation strategies that can be used to solve the conflict. Once the regulation strategies have been used, the participant can select from different responses for the conflict, although only one will be assertive (and thus, correct).

All the conflicts included in this gamified tool have been worked on with the participants. Some examples of them are blaming a teammate, insulting the opposing team, badmouthing a teammate, wetting a teammate’s gym bag, breaking glass, fouling, blaming the coach, laughing at the goalkeeper, and not passing the ball.

In light of the above, and with the hope of offering a tool to improve coexistence, wellbeing, and positive emotions in sport, this paper aims to compare the levels of the variables satisfaction, emotion management, and bullying, in a group of sports students that have been trained with Happy Sport and another group that has not received any treatment.

The value of this research lies in its contribution to the assessment of emotional learning in a sports context and in the context of the 8- to 12-year old population, bearing in mind that the results could contribute to optimizing and directing emotional education programs.

## 2. Materials & Methods

### 2.1. Participants

The final study sample consisted of 194 athletes with a mean age of 11.89 years and a standard deviation of 1.36. Of this sample, 177 (91%) of the participants were boys and 17 (9%) were girls. The lopsided nature of the sample in terms of gender may be due to the fact that the sport chosen for the study was soccer, which, even today, is played by more boys than girls. All the participants belonged to the *benjamín* and *alevín* categories (i.e., 3rd- to 6th-grade primary school students). The experimental group consisted of 65 students from 6 sports schools (34% of the total). The control group consisted of 129 students from 1 sports school (66% of the total). Despite the participation of 6 schools in the experimental group, the students that showed interest in taking part in the study were 65 and not more, as was initially expected (see Table 1). This is due to the fact that the study was carried out with an accidental sample, given its social nature.

However, it is important to note that the groups have been selected in the most equivalent way possible. The assignment of the participants to the groups and their assignment to the program has been carried out randomly to guarantee the equivalence of the groups.

The selection of the population of interest has been made up of a sample of participants belonging to the same age group who are residents in a homogeneous geographical area (the same autonomous community) belonging to a homogeneous social group that are enrolled in public educational centers and participants in sports schools of the same modality.

### 2.2. Design and Procedure

Prior to the start of the study, the participants’ families were contacted and provided with a specific brochure about the study. The study, at all times, complied with the ethical considerations for research projects set out in the Declaration of Helsinki (World Medical Association, Seoul, Korea, 2008), as well as with Spanish law on clinical trials (Law 223/2004 of 6 February 2004), biomedical research (Law 14/2007 of 3 July 2007), and participant confidentiality (Law 15/1999 of 13 December 1999).

The method chosen for the study was a quasi-experimental design with a pre-test and post-test and had a control group. All participants completed the full protocol with self-report tests and were observed at baseline by means of systematic observation. Subsequently, only the experimental group completed the training with the Happy Sport software. Finally, all participants completed the protocols a second time.

The present study was applied with decision-oriented research that followed an idiographic approach in a natural and ecological context. The context was the football schools that implemented the Happy Sport program. The program could, thus, be followed as a subject of study without the need to manipulate the usual conditions. In order to make it possible, the research team first contacted the school coordinators to present the study and request permission to conduct it. The families and participants were informed about the study’s aims and the instruments to be used, and their participation was completely voluntary and anonymous.

The participants completed the questionnaires in the presence of their coach so that he or she could provide support and ensure they were properly administered.

### 2.3. Instruments

A brief situational test and four questionnaires were applied, which were self-administered under the supervision of the trainers, given the age of the participants.

Situational satisfaction test. Developed ad-hoc. It is a qualitative situational test, elaborated ad hoc from a conflictive situation and contextualized in the sports locker room. The subjects are asked what they feel, what they think, and how they act. The athletes had to resolve three conflict situations described in a situational test. The results, numbered on a scale of 0 to 30 (maximum satisfaction) were evaluated jointly with psychometric instruments. The results show scores for emotional awareness, emotion regulation, and social awareness with a satisfactory reliability, as shown in the article of [34].

Emotion Regulation Questionnaire for Children and Adolescents (ERQ-CA, [35]). The ERQ-CA consists of ten items, divided into two subscales, which correspond to two emotion regulation strategies: cognitive reappraisal (6 items) and expressive suppression (4 items). The items are answered using a 5-point Likert-type scale (1 = strongly disagree and 5 = strongly agree). The reliability indices of the two scales in the moment of the study were 0.809 for the cognitive subscale and 0.645 for the expressive suppression subscale, which can be considered acceptable (see Table 2).

Bullying in Sports Questionnaire [36]. This questionnaire is made up of 19 items, written as affirmations and organized into 4 factors—verbal bullying, physical bullying, social bullying, and psychological bullying—the frequency of which is indicated using a 5-point Likert scale (1 = almost never, 2 = rarely, 3 = sometimes, 4 = frequently, 5 = almost always). The alpha index for the instrument during the study was 0.854, which was considered good (see Table 2).

### 2.4. Statistics

Data for the study was processed using the SPSS Statistics 27.0 (IBM, Armonk, NY, USA) of the software.

Tests of homogeneity and normality were carried out, showing that it was possible to use parametric tests. Also, groups were considered equivalent at the pre-intervention moment of evaluation.

For the analysis of the results, the general linear model was used. Specifically, an analysis of variance (ANOVA and MANOVA) was used for the inter-subject effect, taking into account the randomization of the control and experimental groups.

## 3. Results

First of all, we conducted descriptive analyses to compare participants’ pre-intervention means and standard deviations in the experimental and control group at baseline. Table 3 provides this information.

Moreover, the post-intervention descriptive analyses allowed us to explore how participants from both groups responded to the measures (see Table 4).

When analyzing the differences, the satisfaction results revealed significant differences between those children who had followed the program and those who had not, as shown in Table 5. While the control group’s levels remained unchanged, the levels in the experimental group increased. However, the ŋp^2^ value indicates a fairly small effect size.

Nevertheless, the results of the descriptive analyses do show clear and substantial differences in the satisfaction levels of the experimental and control groups after the intervention, with the experimental participants showing greater satisfaction, as reflected in Figure 1.

The behavior of the variable bullying was similar to the one of satisfaction. As can be seen in Table 6, statistically significant differences were found between the control and experimental groups at baseline—albeit again with a fairly small effect size, which could be explained by several factors, such as sample size. However, no significant differences were found at post-intervention.

As for emotional competencies, due to their nature, the analyses were performed according to the subscales (expressive suppression scale and cognitive reappraisal scale). The expressive suppression scale refers to a rather maladaptive strategy consisting of inhibiting the emotional response. In contrast, cognitive reappraisal is an adaptive strategy that focuses on the antecedents; it appears in the initial stages of emotional activity and gives rise to a reappraisal of the initial impression of the situation [37]. Given that they are two independent subscales reflecting very different strategies, it was decided to analyze them separately (see Table 7).

A comparison of the two scales between the experimental and control groups pre- and post-intervention reveals statistically significant differences. A more detailed analysis of these differences shows that, while the pre-intervention evaluation scores between groups remained stable, they varied considerably in the post-intervention evaluations. In this sense, the cognitive scale showed a significant difference in the post-intervention: the control group obtained a value of 2.80 while the experimental group obtained 2.30 (see Figure 2). In contrast, the expressive suppression scale showed that the control group obtained a mean of 1.46 while the experimental group obtained a 2.04 (see Figure 3). Nonetheless, the effect size for both analyses is small.

## 4. Discussion

Conflicts often arise in children and youth sports experiences, just as they do in other social contexts [2]. Classical studies have already shown that inadequately resolved conflicts can have far-reaching, life-long effects for children’s personality and psychological development [38]. At the same time, recent studies have found that, when well managed and with proper training, these same conflicts can be reconstructive and formative [39].

Character development, leadership, sportsmanship, and achievement orientation do not magically occur simply through participation. These benefits generally follow from competent supervision by adults who understand children and know how to structure programs to provide them with positive learning experiences [38,40].

At present, social and emotional education programs are eliciting growing interest among schools and families and have emerged as a tool, which has already been assimilated, for individual development and wellbeing [41]. However, sports have traditionally been less attentive to these kinds of interventions, which can highly benefit sport students and teams (for example, [25]).

This study has sought to analyze the effectiveness of the gamified program Happy Sport, which was designed to develop and improve emotional competencies in sports as a tool for preventing bullying in this context. Specifically, a quasi-experimental design with a control group was used to analyze the effect of training with Happy Sport on a sample of soccer players. To this end, data were collected on emotional competencies, satisfaction, and bullying pre- and post-training.

The results presented here show statistically significant differences between the participants who followed the program and those who did not. Specifically, the athletes who followed the program reported higher satisfaction levels and a significant decrease in the levels of bullying compared to the baseline and to the control group.

The results of the present study also revealed statistically significant differences in the two scales of the ERQ-CA emotion regulation instrument. After the intervention, statistically significant differences were found for the expressive suppression scale. This fact suggests that participants showed a tendency to regulate their feelings instead of suppressing them, and it also suggests a better adaptation in situations of emotional conflict [42]. The participants of the experimental group did not exhibit a higher level of performance in managing emotions through cognitive strategies than the participants of the control group. Although this finding was not initially expected, existing research shows that cognitive strategies are often slow in change due to their stable characteristics, as well as longer-term strategies, which becomes especially true in younger populations [43]. It is also worth noting that several studies have found that, due to the stable nature of emotion regulation and the need for structural cognitive changes, emotion regulation usually changes more slowly than other emotional competencies, such as emotional awareness, and that these changes take place first at the behavioral and then at the cognitive level (for example, [44]).

These findings follow the line of structured interventions that have been analyzed in the context of formal education, such as primary schools or other formal education institutions, and holds hope for improving performance, wellbeing, and assertiveness among students [24,45,46]. Studies like the present one, which evaluated this same effect in non-formal and sports contexts, are less common but equally needed. With this study, we hope to shed some light on the relation between and improvement of emotional competencies, assertiveness, and wellbeing in sports.

## 5. Conclusions

This study aims at extending the knowledge that is needed to improve child athletes’ social and emotional competencies in order to enhance coexistence and satisfaction through a gamified pedagogical tool.

The present work, however, has some limitations that should be taken into account.

First, the study sample was relatively small and had a significant gender bias. Gender differences can be explained by the cultural connotations of the type of sport (soccer) taught at the schools that participated in the study. The sample size may be due to the fact that the sample was accidental and from a region that is not that populated (Pyrenees). Future research should try to include more gender-equal sports.

Additionally, the effect sizes obtained were not high. Although the statistical significance does yield results clearly oriented toward differences between the pre- and post-intervention moments and between the experimental and control groups, future studies should consider increasing the sample size to enable an analysis with greater effects.

Nevertheless, and by way of conclusion, the present study seeks to shed light on an increasingly important topic of study, which is not limited to identifying risk factors but rather seeks to work on and cultivate conflict prevention skills in sports contexts through the gamified program Happy Sport.

## Figures and Tables

**Figure 1 ijerph-19-02596-f001:**
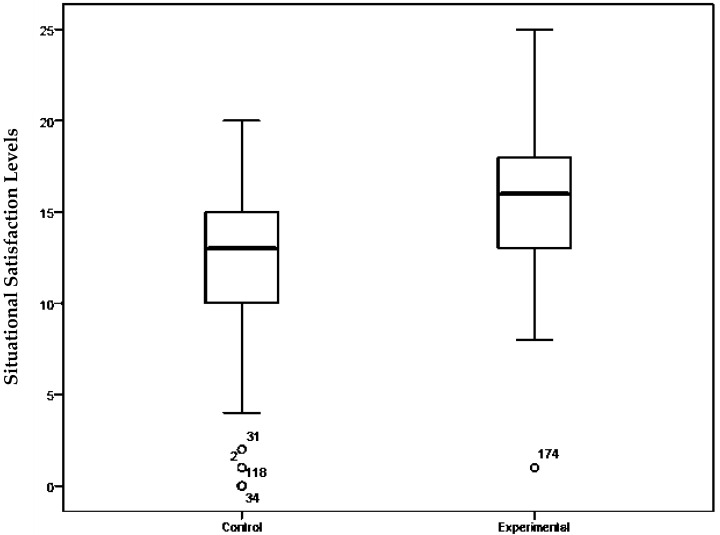
Satisfaction levels of the control and experimental groups in the post-intervention evaluation.

**Figure 2 ijerph-19-02596-f002:**
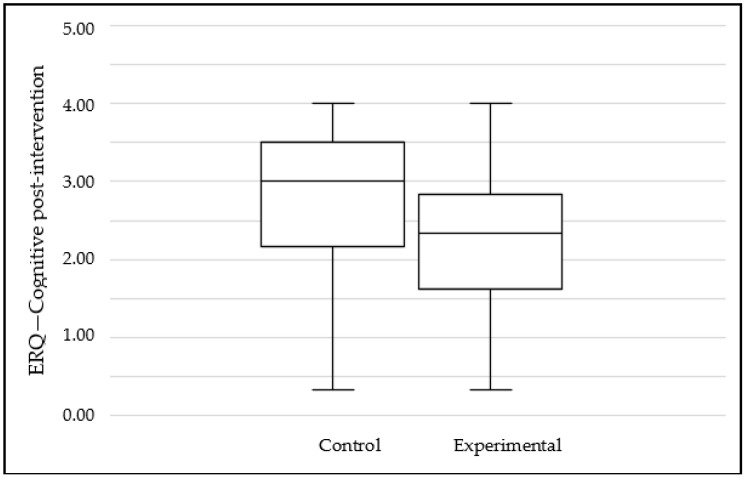
Mean scores for ERQ—Cognitive subscale at post-intervention by groups.

**Figure 3 ijerph-19-02596-f003:**
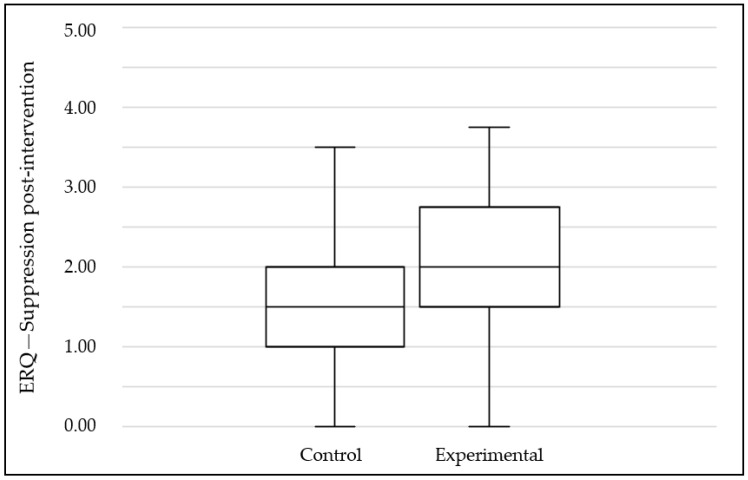
Mean scores for ERQ—Suppression subscale at post-intervention by groups.

**Table 1 ijerph-19-02596-t001:** Description of the study participants.

Participants	N (%)	Age Mean	Age Standard Deviation
Female Male	17 (9%) 177 (91%)	11.81 11.90	1.22 1.37
Experimental group Control group	65 (34%) 129 (66%)	12.16 11.75	1.54 1.23
Total	194	11.89	1.36

**Table 2 ijerph-19-02596-t002:** Cronbach Alpha reliability coefficients for both measures in their two applications.

Measure	Pre-Intervention	Post-Intervention
ERQ-CA—Cognitive Subscale	0.662	0.809
ERQ-CA—Expressive Subscale	0.435	0.645
Bullying in Sports Questionnaire	0.905	0.854

Note: ERQ-CA: Emotion Regulation Questionnaire for Children and Adolescents.

**Table 3 ijerph-19-02596-t003:** Pre-intervention descriptive data for participants in the experimental and control group.

Pre-Intervention	Control Group (N = 129)		Experimental Group (N = 65)	
	M	SD	M	SD
ERQ-CA—Cognitive Subscale	2.02	0.91	2.18	0.74
ERQ-CA—Expressive Subscale	1.85	0.83	1.83	0.85
Bullying in Sports Questionnaire	1.58	0.60	1.29	0.40
Situational Satisfaction Test	12.52	4.39	13.31	3.87

Note: ERQ-CA: Emotion Regulation Questionnaire for Children and Adolescents.

**Table 4 ijerph-19-02596-t004:** Post-intervention descriptive data for participants in the experimental and control group.

Post-Intervention	Control Group (N = 129)		Experimental Group (N = 65)	
	M	SD	M	SD
ERQ-CA—Cognitive Subscale	2.80	0.80	2.30	0.92
ERQ-CA—Expressive Subscale	1.46	0.79	2.04	0.90
Bullying in Sports Questionnaire	1.31	0.34	1.27	0.40
Situational Satisfaction Test	13.23	4.39	15.76	4.64

Note: ERQ-CA: Emotion Regulation Questionnaire for Children and Adolescents.

**Table 5 ijerph-19-02596-t005:** Comparison of inter-subject effects for the variable satisfaction.

	Type III Sum of Squares	df	Root Mean Square	F	*p*	ŋp^2^
Corrected model	187,015	1	187,015	6070	0.015	0.035
Intersection	43,015	1	43,015	1396	0.239	0.008
Differences between groups (pre and post)	187,015	1	187,015	6070	0.015	0.035

**Table 6 ijerph-19-02596-t006:** Comparison of inter-subject effects for the variable bullying at pre- and post-intervention.

		Type III Sum of Squares	df	Root Mean Square	F	*p*	ŋp^2^
Bullying in Sports Questionnaire Pre-intervention	Corrected model	3.626	1	3.626	12.193	0.001	0.060
	Intersection	355.831	1	355.831	11,196.554	0.000	0.862
	Differences between groups (control and experimental)	3.626	1	3.626	12.193	0.001	0.060
Bullying in Sports Questionnaire Post-intervention	Corrected model	0.082	1	0.082	0.627	0.429	0.003
	Intersection	287.437	1	287.437	2185.673	0.000	0.920
	Differences between groups (control and experimental)	0.082	1	0.082	0.627	0.429	0.003

**Table 7 ijerph-19-02596-t007:** Comparison of inter-subject effects for the variables ERQ-Cognitive and ERQ-Expressive at pre- and post-intervention.

		Type III Sum of Squares	df	Root Mean Square	F	*p*	ŋp^2^
ERQ-CA—Cognitive Pre-intervention	Corrected model	1.067	1	1.067	1.452	0.230	0.008
	Intersection	762.900	1	762.900	1038.008	0.000	0.845
	Differences between groups (control and experimental)	1.067	1	1.067	1.452	0.230	0.008
ERQ-CA—Cognitive Post-intervention	Corrected model	11.139	1	11.139	15.646	0.000	0.076
	Intersection	1124.017	1	1124.017	1578.843	0.000	0.892
	Differences between groups (control and experimental)	11.139	1	11.139	15.646	0.000	0.076
ERQ-CA—Expressive Pre-intervention	Corrected model	0.032	1	0.032	0.046	0.830	0.000
	Intersection	584.143	1	584.143	840.786	0.000	0.815
	Differences between groups (control and experimental)	0.032	1	0.032	0.046	0.830	0.000
ERQ-CA—Expressive Post-intervention	Corrected model	14.443	1	14.443	21.049	0.000	0.099
	Intersection	527.509	1	527.509	768.785	0.000	0.801
	Differences between groups (control and experimental)	14.443	1	14.443	21.049	0.000	0.099

Note: ERQ-CA: Emotion Regulation Questionnaire for Children and Adolescents.

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
