# Peer review of "Evaluation of Happy Sport, an Emotional Education Program for Assertive Conflict Resolution in Sports"

_ijerph, 2022, doi:10.3390/ijerph19052596_

Round 1
Reviewer 1 Report
The study is interesting especially for practitioners in sport profession. Below you can find some of my suggestions for improving methodology and would be more informative.
- Explain why number of participants in the experimental and control group is different.
- More info are needed about sample characteristics, especially about equivalence of the groups (see in Conclusion: the sample is ... relatively small and had a significant gender bias.).
- Due to a fact that the Happy Sport program is new approach, more details with some examples would be useful.
- There are some suggestions for Results section:
- Situational satisfaction test is described in 2.3. Instruments section, but doesn't mention after that anywhere. My suggestion is to delete it.
- It is usual to have descriptive statistics for all the applied scales, for every situation, and each group.
- General Linear Model of ANOVA is correct analysis, but interaction part is usually the best indicator of treatment effects.
- After descriptive results would be displayed, there would be no need for present Figures. I suggest to display interaction plots for significant differences.
Author Response
Dear Reviewer,
Thank you so much for all your comments and suggestions.
Please find attached a document with all the changes that we have added in the document.
Many thanks for your help.
The authors.

Reviewer 2 Report
The paper titled "Evaluation of Happy Sport, an Emotional Education Program for Assertive Conflict Resolution in Sports" is an interesting research piece. Well elaborated from the beginning to the end has an exclusive requirements. The graphics information in all figures should be translated from Spanish to English language. Is recommended also to do a final proofreading in the paper.
Author Response

(The authors gave the same response as above.)

Reviewer 3 Report
The paper is very confusing. First, research is conducted in Spain and presumably the authors are not native English speakers, the quality of the English language is not sufficient. My first impression is that the paper needs a thorough proofreading and copyediting. I recommend the authors to have their manuscript reviewed by a native English language speaker.
Abstract: Rewrite according to the guidelines of the Journal.
Introduction:
Summarizes recent research related to the topic. Nonetheless, it is essential to advance the argument/justification about the need for conducting this study. The arguments justifying the need for the study are not well developed, and thereby need to be advanced. In any case, please consider revision, as one long run-on explanation is confusing and difficult to follow.
Use past tense when discussing the procedure and results as well as other researchers' procedures and results.
Method:
The authors reported general Cronbach values dating back to the development of the scale. Is it correct? In any case, it is important to present internal consistency values for the present study.
Change Subjects with Participants
Statistics: The authors should add a separate statistics section. The statistical analyses that were described in the results should be removed and added to the statistics section. The author should give information about what statistics program (e.g. SPSS) was used and whether the statistical requirements for the specific tests were fulfilled (normal distribution etc.).
Discussion:
It would make sense to have the discussion written in accordance with the hypotheses. and in a chronological order.
Pag 1: rewrite references according to the guideline
What is this??? ([4], [5]; [6]; [7]; [8]; [9]; [10]; [11]; etc.) It is wrong!!
Table 1 – Table 2: change Sig. with p
Change Partial eta squared with ŋp2
Author Response

(The authors gave the same response as above.)

Round 2
Reviewer 1 Report
Dear authors,
I try my best to describe in more details my previous suggestions about how to be more clear and precise in your statistical procedures. Also, I send you our paper with similar topics.
- It is usual to have descriptive statistics for all the applied scales, for every situation, and each group.
- The first table(s) in paper must be about means, SDs, and some other descriptive data (e.g. homogeneity and normality) about both samples (exp & control), and in two occasions. To PROVE (as you claim in line 203-205) that the exp and control groups were equal at the beginning of the study, you should compare the group data of all the variables included.
The paper about the same aim is attached. Hope it will help.
- General Linear Model of ANOVA is correct analysis, but interaction part is usually the best indicator of treatment effects.
- When you have control group in your study of the effect of any kind of treatment, then 2 X 2 (cont/exp group X pre/post condition) ANOVA is the best statistical analysis. If there is no significant differences between groups (cont/exp), and the difference is significant after treatment, then the interaction will be significant in favor of exp group.
- After descriptive results would be displayed, there would be no need for present Figures. I suggest to display interaction plots for significant differences.
- You can get interaction only in 2 X 2 (or higher order) ANOVAs. See the paper.

Author Response
Dear Reviewer,
Thank you so much for the clarification.
We have tried to include all the changes, as indicated in our new Cover Letter.

Reviewer 3 Report
no comment
Author Response
Dear Reviewer,
Changes included in the article were sent in the Cover Letter.
Thank you so much for your help and attention.
Round 3
Reviewer 1 Report
You still did not understand and accept majority of the suggestions, even I sent you our paper with similar research design and methodology with clear instruction and explanation how to do. The example of above mentioned is a treatment of a variable called the Situational satisfaction test. There is no even basic information about how it is conceptualized, operationalized and scored, and its descriptive and other important psychometric properties.
Author Response
Dear Reviewer,
Many thanks for your review.
We add a cover letter explaining the changes we have carried out.
Thank you again.
